# Cardiac Asystole at Birth Re-Visited: Effects of Acute Hypovolemic Shock

**DOI:** 10.3390/children10020383

**Published:** 2023-02-15

**Authors:** Judith Mercer, Debra Erickson-Owens, Heike Rabe, Ola Andersson

**Affiliations:** 1Neonatal Research Institute, Sharp Mary Birch Hospital for Women and Newborns, San Diego, CA 92123, USA; 2College of Nursing, University of Rhode Island, Kingston, RI 02880, USA; 3Brighton and Sussex Medical School, University of Sussex, Brighton BN2 5BE, UK; 4Department of Clinical Sciences Lund, Paediatrics, Lund University, 221 85 Lund, Sweden

**Keywords:** asystole, hypovolemia, cord clamping, autonomic nervous system

## Abstract

Births involving shoulder dystocia or tight nuchal cords can deteriorate rapidly. The fetus may have had a reassuring tracing just before birth yet may be born without any heartbeat (asystole). Since the publication of our first article on cardiac asystole with two cases, five similar cases have been published. We suggest that these infants shift blood to the placenta due to the tight squeeze of the birth canal during the second stage which compresses the cord. The squeeze transfers blood to the placenta via the firm-walled arteries but prevents blood returning to the infant via the soft-walled umbilical vein. These infants may then be born severely hypovolemic resulting in asystole secondary to the loss of blood. Immediate cord clamping (ICC) prevents the newborn’s access to this blood after birth. Even if the infant is resuscitated, loss of this large amount of blood volume may initiate an inflammatory response that can enhance neuropathologic processes including seizures, hypoxic–ischemic encephalopathy (HIE), and death. We present the role of the autonomic nervous system in the development of asystole and suggest an alternative algorithm to address the need to provide these infants intact cord resuscitation. Leaving the cord intact (allowing for return of the umbilical cord circulation) for several minutes after birth may allow most of the sequestered blood to return to the infant. Umbilical cord milking may return enough of the blood volume to restart the heart but there are likely reparative functions that are carried out by the placenta during the continued neonatal–placental circulation allowed by an intact cord.

## 1. Introduction

Births involving shoulder dystocia, tight nuchal cords, or occult cords can deteriorate rapidly. The fetus may have had a reassuring tracing just before birth yet may be born without any heartbeat (asystole). In 2009, we published a paper on cardiac asystole based on two cases of infants with unremarkable fetal heart tracings before birth but cardiac asystole at birth [1]. Since that time, three additional publications have described similar occurrences in five infants with acceptable fetal heart rate tracings in the second stage who developed cardiac asystole immediately at birth [2,3,4]. All infants were treated according to current resuscitation protocols at the time which included immediate or early clamping and cutting of the umbilical cord to hand the infant over to the neonatal team for resuscitation [5,6]. In most of the cases, infants experienced shoulder dystocia although the times between the head and body birth were not long enough to cause the amount of damage evident in Table 1 [7]. The cardiac asystole hypothesis suggests that severe hypovolemic shock can lead to cardiac arrest secondary to the loss of fetal–placental blood volume. Immediate cord clamping (ICC) gives the infant no access to the sequestered blood volume in the placenta. Loss of this blood volume due to ICC initiates inflammatory responses which can enhance neuropathologic processes including seizures, HIE, and death [8,9,10,11].

Our purpose is to discuss a probable mechanism for cardiac asystole at birth which highlights the role of the autonomic nervous system (NS). We put forward the need to allow these infants several minutes with an intact cord for the blood to return from the placenta before the cord is clamped. We propose an algorithm to accomplish this goal which differs from current guidelines.

## 2. The Cardiac Asystole Hypothesis

The original hypothesis, first published in 2009, has not changed and has been supported by additional publications with similar cases [1,2,3,4]. New information needed to further understand the mechanisms of cardiac asystole is added here. Briefly, the original hypothesis stated that sudden cardiac asystole at birth is due to severe hypovolemic shock. The infant is squeezed in the birth canal causing pressure on the umbilical cord. The firm-walled umbilical arteries are not compressed and can continue to transfer blood from fetus to placenta. However, the soft-walled umbilical vein cannot return this blood to the fetus as it is more easily compressed. During the second stage of labor, the heart rate remains normal because pressure from the vaginal walls act as an anti-shock garment and supports the transfer of blood from the peripheral circulation to the central circulation [12]. This pressure maintains heart rate and blood pressure to the most vital organs. At birth, there is a sudden release of the vaginal wall pressure (akin to quickly removing an anti-shock garment) allowing blood to flow rapidly into the peripheral circulation and reducing central circulation. The decrease in central circulation reduces cardiac preload and can lead to severe bradycardia and even asystole as demonstrated in the cases in Table 1. Clamping the cord immediately prevents access to the sequestered blood and leaves the infant, with only its fixed blood volume available at birth, appearing very pale (Figure 1). If the heart is restarted after asystole and ICC, the reduced blood volume may stimulate inflammation resulting in hypoxic–ischemic damage throughout the body [8,9,10]. We believe that the original hypothesis is accurate and complete except for providing the pathophysiologic mechanisms for why the heart stops suddenly. This new part of the theory is elaborated in the following paragraphs. 

## 3. The Cases 

Table 1 displays the commonalities among published cases of infants who experienced cardiac asystole at birth. Note that all had acceptable fetal heart tracings during the second stage of labor, yet all had asystole at birth. All had ICC and time to first heartbeat ranged from 7 to 25 min. One newborn could not be resuscitated and died. All had full resuscitation with chest compressions, intubation, and epinephrine several times. In three of the cases, the hearts restarted within one minute after a large dose of saline (14 to 30 mL/kg), but two infants did not respond to smaller doses (~10 mL/kg). The fact that three infants’ hearts restarted within one minute after a larger dose of saline, lends support to the theory that fluid replacement with the infant’s own blood volume will restart the heart rapidly [13]. Initial cord pH levels were above 7 but, within one hour or less, most were 6.7 to 6.9. All surviving neonates were diagnosed with HIE. Four died and the other three had developmental delay and/or cerebral palsy.

## 4. Evidence of Benefits of Placental Transfusion 

A placental transfusion (after 3 min at the level of the perineum) provides a term neonate with approximately 80 to 100 mL of fresh whole blood [14,15,16]. Receiving this blood results in a 3- to 12-month supply of iron resulting in higher hemoglobin and hematocrit levels initially and less anemia and iron deficiency in infancy [17,18,19,20]. Improved myelin volume in the developing brain was documented at 4 and 12 months [21,22]. Rana reported better communication and personal social skills at 12 months after a 3-min delay in cord clamping [23] while Andersson found better motor and social development at 4 years of age especially for boys [24]. For preterm infants, a placental transfusion of only 30 to 60 s resulted in decreased mortality by 30% in the NICU and out to two years of age [25,26]. Fewer deaths were accompanied by less intraventricular hemorrhage, fewer gastrointestinal issues, and lower transfusion requirements [27,28,29]. No harm was noted to mothers in any of the studies. Winkler reported less postpartum hemorrhage with extended delayed cord clamping (6-min average) among 9000 women delivered by Swedish midwives [30].

For term infants needing resuscitation, Andersson et al. (2022) found higher 5- and 10-min Apgar scores after a 3-min delay in cord clamping and better neurodevelopment at two years of age compared to those infants who had ICC [31,32]. In a pragmatic cluster-randomized crossover trial of cord milking (four times) for non-vigorous infants compared to ICC (*n* = 1730), Katheria et al. reported less need for cardiopulmonary resuscitation, a lower incidence of HIE, and less need for therapeutic hypothermia for infants in the milking group suggesting that additional blood volume at birth is a possible mode of prevention for HIE [33]. This is the first trial to show that umbilical cord milking is safe and superior to ICC for term newborns in need of resuscitation. 

The duration of the delay until the time of cord clamping appears to impact the infant’s return to stability. In a resuscitation study using near-term newborn lambs, Polglase et al. found that a 10-min delay in cord clamping prevented post-asphyxial rebound hypertension which was seen in the lambs who had either ICC or a 1-min delay in cord clamping [34]. Polglase stated that “the timing of umbilical cord clamping after return of spontaneous circulation, is critically important for improving physiological stability particularly blood pressures, flows, and oxygen delivery to the brain.” This important finding suggests a mechanism of action for causation of intraventricular hemorrhage and other damage to the neonatal brain. It is known that the placenta aids the fetus in the clearance of catecholamines during pregnancy [35,36]. The Polglase et al. study raises the question of whether that role (and others) continues after birth while the cord remains intact. No harm to human or animal infants was reported in any of the studies mentioned. Excellent recent reviews are available [37,38].

### 4.1. Basic Physiology of Placental Transfusion 

In utero, about 2/3rds of the fetal placental blood volume flows through the term infant’s body at any point in time and 1/3rd is circulating through the placenta [39]. Thus, delayed cord clamping (DCC) has the potential to add up to 30% of a newborn’s blood volume improving hemoglobin, birth weight, and hemodyanmic stability [40]. For the smallest preterm infants, the amount of blood in the placenta approaches 50% [41]. This blood is not “extra”—it is the blood that the fetus uses to obtain essentials from the placenta and eliminate waste products. The fetal heart pumps this large amount of blood out through the umbilical cord, through the placenta, and back through its body. With ICC, a large amount of blood remains in the placenta and is unavailable to the newborn [39,42]. An intact cord allows continuous flow into the right ventricle which increases pulmonary blood flow [43,44]. 

### 4.2. Important Contents of Cord Blood 

To understand the impact of acute hemorrhage on the newborn, it is important to consider the unique contents of cord blood. Warm (body temperature), oxygenated, residual placenta blood is readily available to the infant whose umbilical cord remains intact providing placental circulation and respiration for several minutes after birth. This blood offers about 15–20 mL/kg of red blood cells—enough to provide the term newborn additional oxygen-carrying capacity and adequate iron for up to 12 months [38,45]. A total of 15 to 20 mL/kg of plasma carries many essential factors that support and drive the process of transition [41]. Plasma also provides the perfect medium for the millions of stem cells providing an autologous transplant to support the newborn’s immune system [46]. Lawton et al. suggested that these stem cells may reduce the neonate’s susceptibility to neonatal and age-related diseases [47]. Neuroprotective progesterone in the term neonate’s blood at birth is almost two-times higher than the mother’s level [48]. It likely causes the vasodilation essential to distribute the large amount of placental transfusion throughout the body as all organs must now function independently without the placenta [49,50]. Vasodilation should help to prevent ischemia. The enhanced vascular perfusion provides higher pulmonary artery pressure for several hours. In a study using cardiac catheterization on normal term infants in 1966, Arcilla et al. showed that the pulmonary artery pressure remains higher for at least 10–12 h in infants with DCC versus those with ICC [51]. Higher pressures from enhanced perfusion provide mechanical stimuli (mechanotransduction) which causes electrochemical signaling to the endothelial cell linings. This stimulates endothelial cells to release vital angiocrine messengers that are essential for normal function, maturation, maintenance, and repair of all organs [50,52].

### 4.3. Effect of Blood Loss at Birth on the Fetus/Newborn

The effect of blood loss at birth from ICC can be quite subtle or acute [53]. In a study examining the effects of a severe acute hemorrhage on late-term fetal sheep, Meyers and Rudolph (1991) found life-threatening effects following the removal of 40% of the fetal blood volume [54]. Although the lambs were able to restore blood pressure and heart rate (from hemorrhage-induced bradycardia), they could not maintain oxygen delivery to the tissues resulting in a fall in oxygen consumption and rising metabolic acidosis. Removal of more than 40% of the fetal blood volume resulted in death of the near-term lamb fetuses [54]. While about 30% of the human fetus’s blood volume is in the placenta at any point in time, fetuses with cord compression may have much more blood sequestered in the placenta. This puts those infants with ICC at higher risk of poor circulation and oxygen delivery after birth leading to ischemia, hypoxia, and even death.

### 4.4. The Importance of Unconsciousness and Lack of Tone

All the infants in Table 1 appeared unconscious at birth accompanied by a lack of breathing, tone, and reflexes. The most common reason for an adult or newborn to lose consciousness is loss of blood flow to the brain and brainstem [55]. Impaired perfusion to the brainstem results in unconsciousness as well as poor or no muscle tone or reflexes. The pontomedullary reticular formation in the brainstem is best known for promoting arousal and consciousness. All the cranial nerves pass through it [56,57], Figure 2. Wijdicks refers to the brainstem as the “seat of consciousness“ [58]. Reticular formation neurons form circuits with the motor nuclei of the cranial nerves responsible for the movements and reflexes of the face, head, and neck [59]. The fibers from these pathways ascend superiorly to the upper brain centers (thalamus, amygdala, cerebrum) to promote wakefulness, arousal, and vigilance and inferiorly to the viscera (heart and lungs). This pattern constitutes a “first-alert” communication network reaching the entire body through the Reticular Activating System (RAS) and the autonomic nervous system (Figure 1). Reduction in blood flow to this area results in unconsciousness, areflexia, atonicity, and apnea as seen in all seven case infants in Table 1.

### 4.5. Notes on Breathing

The prevailing belief is that the newborn lung, if the infant is not breathing, must be opened with high initial negative pressure which will decrease the pulmonary vascular resistance and push lung fluid out of the alveolar spaces in order to prevent hypoxia [60]. However, an intact cord can provide placental circulation and respiration as evidenced by the newborn’s heart rate and improving color. It supports the infant while perfusing the body and brain with blood like what happens in the ex-utero intrapartum technique (EXIT) [61,62].

Lang demonstrated that severing the vagus nerve in rabbit fetuses prevented the previously seen influx of pulmonary blood flow with assisted breathing [63]. When the vagus nerve is intact and well perfused, it likely instigates the shift in the alveolar membrane from making lung fluid to excreting it [35,64,65]. Intact cord resuscitation provides oxygenated blood flow and time to fully perfuse the brainstem where the neurons driving the reflex circuits of the autonomic nervous system are contained [66].

Recent studies have reported that applying a face mask or even nasal prongs to a newborn immediately after birth may inhibit initial breathing and reduce the heart rate [67,68,69]. Applying these devices induces vagally mediated facial reflexes that inhibit spontaneous breathing. The trigeminocardiac reflex and the laryngeal chemoreflex can also be elicited by air flow, leading to glottal closure [70,71] This response is like the diving reflex which displays dramatic power over autonomic function causing apnea, bradycardia and vasoconstriction, and altering normal homeostatic reflexes such as the baroreceptor reflex and respiratory chemoreceptor reflex [66].

There are reports of reduced blood flow to the heart with breathing movements. Newborn lamb experiments found that the umbilical venous flow was markedly reduced with each breath [72] and that a 30-s sustained inflation, even with delayed cord clamping, prevented the lamb’s blood from flowing into the lung via the pulmonary artery and inferior vena cava [73]. A randomized control trial comparing sustained inflation with usual ventilation in human preterm infants was stopped because more early deaths occurred in the group of infants with sustained inflation [74]. These findings demonstrate that early or forced breaths may interfere with a physiologic transition by blocking or slowing the essential increased blood flow to the heart. Aerating the lung is critical for the success of neonatal transition but doing so before the lung and brain have been adequately perfused may be harmful. A heart rate above 100 or increasing indicates wellbeing and continued placental respiration [6]. Thus, attempting to start ventilation immediately after birth, and before adequate perfusion especially of the brainstem, may compromise initial and early breathing as well as transition [75]. It is much easier to push air into lungs when the alveolar capillary circulation is fluid-filled [76,77]. Both Katheria and Nevill reported that for preterm infants, there is no harm in waiting 60 s after birth to deliver the first assisted breath [78,79]. They found that 90% of the infants had breathed on their own by 60 s and that no clinical benefit was reported from assisted breathing before that time.

## 5. The Autonomic Nervous System: Fetal-Newborn Implications

The autonomic NS, through neural centers in the brainstem, maintains our wellbeing as it coordinates control over vital systems throughout our lifetime including the fetal and newborn periods [80]. At birth, and for the newborn, the physiological adaptive systems of the autonomic NS provide brisk and sustained responses to aid in the cardiovascular system’s adaptation to life without a placenta. It is important to note that the autonomic NS does not require conscious awareness to function [59]. All vertebrates have an internal surveillance system that can sense signals of danger. In mammals it extends to awareness/detection of signs of safety. Sensing whether they are in a safe environment allows mammals to have positive social interactions essential for group living and survival [81]. The mammalian ability to sense safety prevents non-essential sympathetic activation and thus protects the oxygen-dependent central NS and overall health [82,83]. Early adverse events, even prior to birth and during the neonatal period, can impact maturation and function of the autonomic NS and can cause dysfunction in essential adaptive responses leading to disease [80,84].

The autonomic NS is often described as having two major circuits—the sympathetic (flight or fight) and the parasympathetic (rest and digest) systems. In mammals, the parasympathetic system has two functionally distinct vagal pathways: [59,83] a primitive dorsal vagal component and a phylogenetically newer ventral vagal component as shown in Figure 3. These three pathways are vital to understanding the cardiac asystole theory and are discussed below.

### 5.1. The Autonomic Nervous System: Three Pathways

The three preset pathways (sympathetic, ventral vagal, and dorsal vagal) of the autonomic NS (Figure 3) work in a specific hierarchical order in the body in relation to events that affect our daily lives [85]. Briefly, the ventral vagal circuit predominates creating a calm state that supports health, growth, and restoration along with social connection and interaction. When challenged or frightened, one moves to a mobilized defensive fight or flight state due to activation of the sympathetic NS. Under severe threat, activation of the most primitive dorsal vagal pathway leads to immobilization such as behavioral shutdown, withdrawal, vasovagal response, syncope, and death-feigning [59,83]. Each pathway is described in more detail below.

#### 5.1.1. The Sympathetic Nervous System

The sympathetic NS is best known for the stress response and prepares one for fighting or fleeing. With sympathetic NS arousal there are multiple immediate massive physiological responses which disrupt digestion, increase heart and breathing rates, heighten hearing acuity, dilate pupils, blast all energy to the skeletal muscles, brain and heart, and release catecholamines. The sympathetic NS is the key player in the physiology of stress even when we are not in immediate danger [82,86]. The positive side of the sympathetic NS supports energetic activities and prevents us from being lethargic. It provides an energy source for our exuberance, liveliness, and good feelings when not supporting flight or fright responses. It works in concert with the ventral vagal nerve until activated by real or assumed danger when it predominates, and the ventral vagal complex is inactivated.

#### 5.1.2. The Ventral Vagal Complex

The ventral vagal complex (VC) which arises from the nucleus ambiguus in the brainstem has evolved to be the predominant vagal system for mammals and fosters psychological and physiological health (Table 2). It rapidly and efficiently regulates visceral organs including cardiopulmonary functioning [83]. It provides neural influences for the sinoatrial node (the cardiac pacemaker) via the release of acetylcholine which inhibits cardiac activity and attenuates the influences of the sympathetic NS. Without this inhibition, the heart would beat 90–100 times per minute, but the ventral vagus nerve keeps the normal heart rate at ~70 beats per minute unless it is inactivated by the sympathetic NS [83]. There is also an anatomic and neurophysiologic link between neural regulation of the heart, via the ventral vagus, and the special visceral pathways that regulate the muscles of the face, head, and neck. This results in a face–heart link (social engagement system) for mammals which has important influence on newborn transition and the co-regulation of communication between mother and infant essential for survival [59]. Thus, in addition to its influence on the heart and other functions, the ventral VC is associated with feelings of safety and displaying spontaneous affiliative social behavior [83].

#### 5.1.3. The Dorsal Vagal Complex

The dorsal vagus complex (dorsal VC) first evolved in our reptilian ancestors. This pathway continues its role to support health, growth, and restoration via neural regulation of the subdiaphragmatic organs in mammals (Table 2). Usually, it works smoothly with the ventral VC and the sympathetic NS as it regulates digestion and absorption. However, when a life-threatening event occurs, the other two pathways are likely inhibited and the dorsal VC is activated resulting in a passive defensive strategy that may lead to immobilization or death (e.g., freezing, vasovagal fainting, death-feigning, and asystole) [83,87].

#### 5.1.4. Responses to Life Threatening Events

Fear and/or life-threatening events activate the two defense systems: the sympathetic NS and the system of immobilization and disassociation linked with the dorsal vagal pathway [88]. When the higher neural system arrangements (VVC and SNS) are suddenly rendered functionless, the lower (older dorsal vagus) becomes active. Using unconscious awareness or “neuroception” (in contrast to perception which does require awareness), the body (including that of a fetus or newborn) can distinguish environmental and visceral features that are safe, dangerous, or life threatening and responds appropriately [83,89]. When in a safe environment, the autonomic state is regulated by the ventral VC which dampens sympathetic activation and protects the oxygen-dependent central NS from the sympathetic NS and the dorsal VC’s metabolically conservative reactions. When the body is challenged with a fearful event, the ventral VC allows the sympathetic NS to dominate. If the event is life-threatening (such as a severe hemorrhage), both the ventral VC and the sympathetic NS give way to the dorsal VC. The dorsal VC sends inhibitory impulses to the heart to conserve metabolic resources by reducing cardiac output. This inhibition constricts the bronchi and inhibits the cardiac pacemaker and is most likely responsible for the apnea and bradycardia seen in preterm infants [90]. It is also the most likely explanation for the asystole seen in the seven case infants found in Table 1.

### 5.2. The Mechanism for Asystole due to Severe Hypovolemia

In the presented cases (Table 1), it is likely that cord compression resulted in a large quantity of the fetal–placental blood volume becoming sequestered in the placenta causing hypovolemic shock in the neonatal body. Strong vaginal wall pressure during the second stage of labor acted like an anti-shock garment maintaining fetal central circulation and resulting in a near-normal or normal fetal heart tracing negating evidence of in-utero hypoxia and ischemia. The sudden release of pressure at birth allowed the infant’s already low blood volume to move from the central circulation to the peripheral circulation resulting in a severe reduction in central blood flow. The sequestration of blood in the placenta created the equivalent of a sudden severe hemorrhage similar to what Meyers caused in the lamb fetuses [54]. ICC prevented any blood from the placenta to return to the asystolic infants. This situational hypovolemia resulted in preload failure reducing blood flow to the right atrium alerting the sinoatrial node and other baro- and chemoreceptors to the lack of adequate blood volume. These events were so critical that they likely caused the infants’ hearts to stop beating immediately influenced by the dorsal vagal complex [91]. It is important to note that the dorsal vagus acts on the heart only when the other parts of the autonomic NS fail [83]. The most likely cause of immediate cardiac asystole in the case infants was due to the response of the dorsal vagus to the life-threatening event of acute sudden massive loss of blood sensed by the body after release of the vaginal wall pressure.

### 5.3. Cause of Death

Immediately clamping the cord of pale, non-vigorous infants, who likely have blood sequestered in the placenta, may increase the likelihood of adverse outcomes including disability and/or death. ICC prevents a life-saving placental transfusion. In the reported cases (Table 1), all infants had ICC. It is likely that when the pulseless infant remains attached to its umbilical cord (and placenta), it will receive its placental transfusion via the release of pressure on the intact umbilical cord [13]. Even if the infant’s heart is asystolic, pressure from the overdistended placenta and the uterus will cause blood to flow into the infant. Return of the blood volume to the newborn’s body, dispersed to the heart, brainstem, and brain, allows consciousness, tone, reflexes, and vision to return all together [13,92]. The sudden reoccurrence of the brainstem reflex reactions simultaneously indicates adequate perfusion of the reticular activating system (Figure 2), brainstem, and brain likely including the watershed areas [58].

## 6. Clinical Implications: Preventing Death from Asystole at Birth

The birth of a pale and toneless newborn presents a critical situation at delivery and creates a fearful experience for all involved, especially after the stress of managing a shoulder dystocia case. The instinct of an obstetrical provider is to immediately clamp and cut the cord and rapidly passing the limp infant to a nurse or the resuscitation team if present. While the current resuscitation guidelines allow one minute for warming, drying, stimulating, and assessing, often that time is bypassed with a pale, non-vigorous newborn and the cord is clamped immediately. However, it is likely that immediate cutting of the umbilical cord ensuring the loss of the placental blood volume intended for the newborn, may result in disability or death [13].

The cardiac asystole hypothesis stresses the importance of keeping the umbilical cord circulation intact during the first several minutes of life for non-vigorous infants (Figure 4 Algorithm). This accommodates physiologic neonatal transition for all newborns and especially for those most compromised infants [93]. The fresh whole blood the infant can receive if the cord is intact prevents ischemia and hypoxia [10]. In contrast saline, often given to bolster blood pressure in a hypovolemic infant, only dilutes the blood and is soon excreted. Alternative practices to ensure an intact cord for these infants requires education about the danger of ICC, a plan for intact cord resuscitation for non-vigorous infants, preplanning, and practice (i.e., drills) for the whole team [94,95].

Figure 4 represents an alternative algorithm we propose for management of a non-vigorous, likely hypovolemic infant who appears extremely pale at birth without tone, breathing, or other reflexes. Autonomic brainstem responses (consciousness, breathing, reflexes, tone) are all suppressed due to poor brainstem perfusion from the hypovolemia. Key to the successful management of this infant is to support the return of sequestered blood volume from the placenta to perfuse all parts of its brain and body. Steps in the algorithm are discussed below.

### 6.1. Where to Place the Infant

One of the important barriers to intact cord resuscitation with a non-vigorous infant, is the traditional hospital delivery bed with the lower section removed and the birthing person’s legs up in stirrups. Several moveable tables have been developed to aid in resuscitation used mainly with preterm infants [37]. These tables are generally costly, and few units can afford enough for each delivery room. However, we suggest an early work-a-round. When using a labor bed with the lower section removed, the easiest method to ensure warmth and time for placental transfusion, is to cover the baby quickly in a warm blanket and place it on the mother’s abdomen. Gentle stimulation can be provided thru the blanket and the cord can be seen near the mother’s perineum. If there is an assistant available, that person can put back the lower section of the bed and remove stirrups if needed. If there is no assistance, leave the neonate at the mother’s abdomen, call for help, and proceed to put back the lower section of the bed. This is usually possible to perform within 30 s (see SAVE film in Swedish at https://youtu.be/AlV4D-UvOS0?t=201, accessed 27 January 2023). If the infant responds well—that is the heart restarts and the infant cries—it can stay on the mother’s abdomen. If the neonate is still non-vigorous, proceed as follows, which also applies to birth on an unbroken bed. Place a clean pad at the perineum and position the infant on the pad covered with a warm blanket. As the provider dries and provides gentle stimulation, the umbilical cord flow can be assessed.

### 6.2. Assessing Heart Rate

Observing or gently checking the cord for pulsations will indicate if there is good blood flow. It is likely that if there is a large amount of sequestered blood, the flow will start immediately upon release of the pressure on the cord due to the high pressure in the placenta and uterus because of blood backup. This can be verified by cord pulsations and heart rate. To confirm heart rate, one can check the cord (gently) or, even better, use a doppler to listen to chest [96]. Use of a doppler on the infant’s chest allows everyone in the room to hear the heart beating which should help reduce stress and panic among staff especially while the change from ICC and rushing the infant away from the mother to intact cord resuscitation attached to the mother via an intact cord is underway. If the cord is unobstructed (and especially when the infant is at or below the placenta), the heart of the asystolic infant should start beating in 15 to 20 s as the heart fills with blood stimulating the AV node, baroreceptors, and other circulatory sensors [97].

#### 6.2.1. Positive Heart Rate Response

As soon as the heart starts beating again or the rate increases, one can assume that the placenta is providing the newborn with the same level of oxygenation that was available in utero and the infant’s color will improve as the skin is better perfused. The infant will likely still be apneic, atonic, and unconscious due to under-perfusion of the brainstem where these reflexes initiate. The infant’s beating heart will provide more blood volume to the brainstem (and the entire body), but it can take a few minutes to distribute the residual placental blood volume.

#### 6.2.2. Poor Heart Rate Response

If there are no pulsations after 15 to 20 s or the heart rate is less than 100, positive pressure ventilation can be started. If that does not start or increase the heart rate, the cord can be milked 4 to 5 times providing the infant with a bolus of blood [33]. If that restarts the heart, then the cord should still be left intact for several minutes more to allow the placenta to help the infant recover from the stress of birth [13,31,98,99]. If milking the cord does not start or increase heart rate above 60, then a full standard resuscitation needs to be started [6]. In this case, whether chest compressions alone would restart the infant’s heart and blood flow is unknown. It is preferable to conduct a full resuscitation with the cord intact if one has the capabilities.

### 6.3. Brainstem Reflexes: Consciousness, Breathing, and Tone

After the infant has a good heart rate and color begins to improve, return of brainstem functions (breathing, tone, reflexes) may still take a few minutes as the infant needs to fully perfuse the brainstem and the brain. That brain stem functions return all at once confirms that poor perfusion (resulting in poor oxygenation) was the cause of the malfunction [13]. However, giving oxygen without good perfusion may not revive the infant and may result in additional oxidative stress [34,78]. Cutting the cord of the hypovolemic infant (Table 1) to resuscitate away from the mother may result in death, HIE, end-organ ischemia, and inflammatory issues [8,9,10].

### 6.4. Skin-to-Skin Care of the Infant

After brainstem functions return, it is important to place the infant on the maternal abdomen and leave the cord attached for the continuation of placental circulation for several more minutes to allow the infant to fully recover from this stressful birth and allowing for care there [99,100]. One benefit of keeping the low-resistance placental circulation open is that it may avoid the hypertensive overshoot identified by Polglase [34]. It may also aid in calming the infant by providing assistance from the placenta in lowering catacholamines [36].

The infant and mother can be observed together allowing for skin-to-skin contact between them. This is an important step for an infant who has been quite stressed at birth. Benefits of skin-to-skin (STS) care are so numerous that it should be top priority for the recovering and all newborns [99,101]. Some providers will want to send the recovering newborn to the NICU or nursery for “observation”, but observation should take place on and with the mother to reap the many benefits of skin-to-skin time for both the mother and infant.

Throughout this process, watching for signs such as diameter and fullness of the cord, cord pulsations, continued good heart rate, improving color, infant movement, and tone (flexing, moving), consciousness (eyes open, will search for mother’s/parents faces), will reassure all that the newborn is recovering fully. Leaving the cord intact until the newborn is conscious, responsive, fully perfused, and the placenta is ready to deliver or has delivered, is the wisest choice for infant health and well-being.

## 7. Research

While resuscitation with an intact cord is routine in out-of-hospital settings (home and birth center births) in the USA and Canada, few in-hospital providers have attended such births [92,93]. Many midwives delayed clamping the cord initially so that no one could remove the infant from skin-to-skin contact with the mother. In a recent survey, 54% of midwives reported that one of the main barriers to practicing DCC was time pressures to hand off the newborn [102]. They also found it challenging to practice DCC in situations when the newborn needed resuscitation although they found DCC beneficial in out-of-hospital settings [93]. Yet, historic references of intact cord resuscitation are found as early as writings of Aristotle [103].


*“It often happens that the child appears to have been born dead when it is merely weak, and when before the umbilical cord has been ligatured, the blood has run out into the cord and its surroundings. But experienced midwives have been known to squeeze back the blood into the child’s body from the cord, and immediately the child that a moment before was bloodless came back to life again”*
In [104]

John Whiteman Williams recommended “late ligation” based on the evidence from a colleague who demonstrated that 92 ccs of blood could be recovered from the maternal end of the cord after early ligation indicating that that a large amount could be lost to the infant by early clamping). This simple informative method of documenting the absence or presence of residual placental blood volume is valuable but too rarely used. Large collections of harvested cord blood validate this point [105]

As the need for resuscitation for term infants is often unanticipated before birth, research involves enrolling large numbers of women before or during labor. Two such trials are described here. These two important trials should assist in changing the existing opinion away from ICC for non-vigorous infants and instead provide delayed cord clamping or cord milking. A multi-center randomized controlled trial in Sweden, known as the Sustained cord circulation and Ventilation study (the SAVE-method) is underway (NCT04070560) [95]. The research team anticipates screening 8000 births to obtain 600 late preterm and term neonate subjects in need of resuscitation [95]. The team has completed phase 1 which included developing a method of providing in-bed intact cord resuscitation. In phase II, they adapted the method for use in multiple settings. Phase III will involve the initiation of a full-scale multicenter study. Throughout the development of their protocols, they have continued to identify barriers and facilitators for intact cord resuscitation.

Another important study is The Milking in Non-Vigorous Infants (MINVI) trial compared milking with ICC for infants needing resuscitation [33]. This international multi-center trial (*n* = 1730) used a cluster randomized crossover study design and was successful in obtaining waivers of consent allowing for randomization of hospitals rather than individuals (NCT03631940). Each hospital was randomized to either cord milking or ICC for one year and then switched to the other treatment for a second year. The primary outcome was admission to the NICU while HIE was the main safety outcome. UCM did not reduce NICU admissions significantly but there were fewer cases of HIE in the experimental group. Infants in the UCM group also had higher 1 min Apgar scores and higher hemoglobin levels. Results of two-year follow up using parent assessment tools are currently being analyzed. This study is the first to show that the use of cord milking is a reasonable safe option for non-vigorous infants and likely a preventative measure for HIE.

For the study of rare events, case reports, cohort, and epidemiologic studies, as well as quality improvement projects can all lend consideration for generation of hypothesis and successes or failures of current guidelines for care of infants needing resuscitation.

## 8. Conclusions

The cardiac asystole hypothesis is supported by understanding the body’s physiologic reaction to severe situational hypovolemic shock at birth. Asystole is most likely mediated by the dorsal vagal complex as the infant’s body interprets the sudden severe massive hemorrhage as a life-threatening event. Immediately clamping the cord at birth when the infant is pale and toneless, likely ensures death or disability. Umbilical cord milking may return enough of the blood volume to restart the heart but there are likely reparative functions by the placenta during the continued neonatal–placental circulation. Keeping the cord intact facilitates physiologic neonatal transition for infants and is most relevant for infants facing the worst outcomes at the time of birth.

## Figures and Tables

**Figure 1 children-10-00383-f001:**
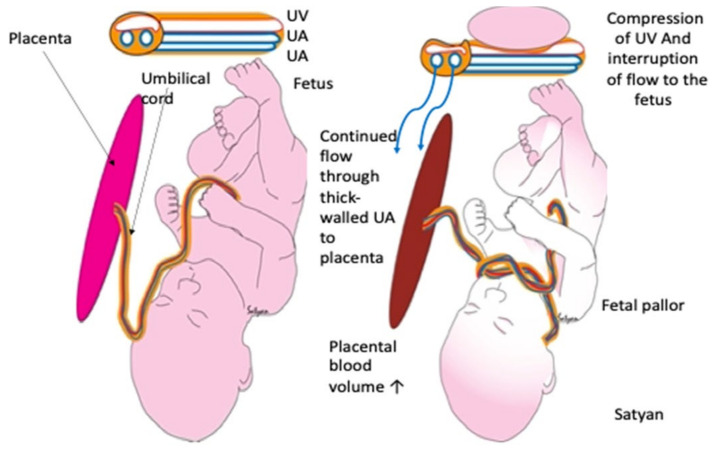
Effects of umbilical cord occlusion on the umbilical vein (UV) and umbilical arteries UA), placenta, and neonate. On the left there is no occlusion; on the right compression of the umbilical vein and interference with the flow from placenta to infant can be seen. Copyright by Satyan Lakshimrusimha, MD. Sacramento, CA, USA. Used with permission.

**Figure 2 children-10-00383-f002:**
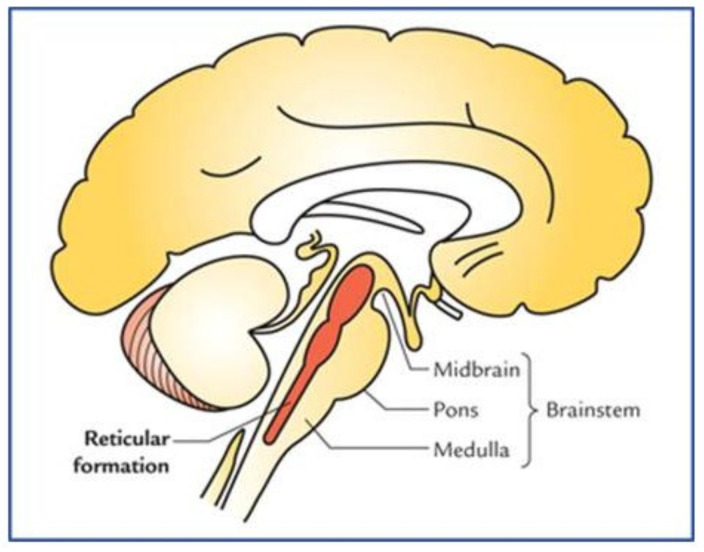
The Reticular Formation [57] (fair use).

**Figure 3 children-10-00383-f003:**
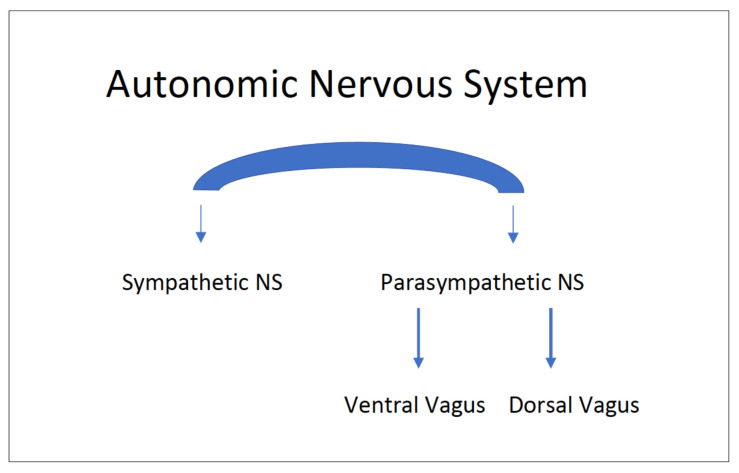
Autonomic Nervous System pathways (Adapted with permission from D. Dana [84]).

**Figure 4 children-10-00383-f004:**
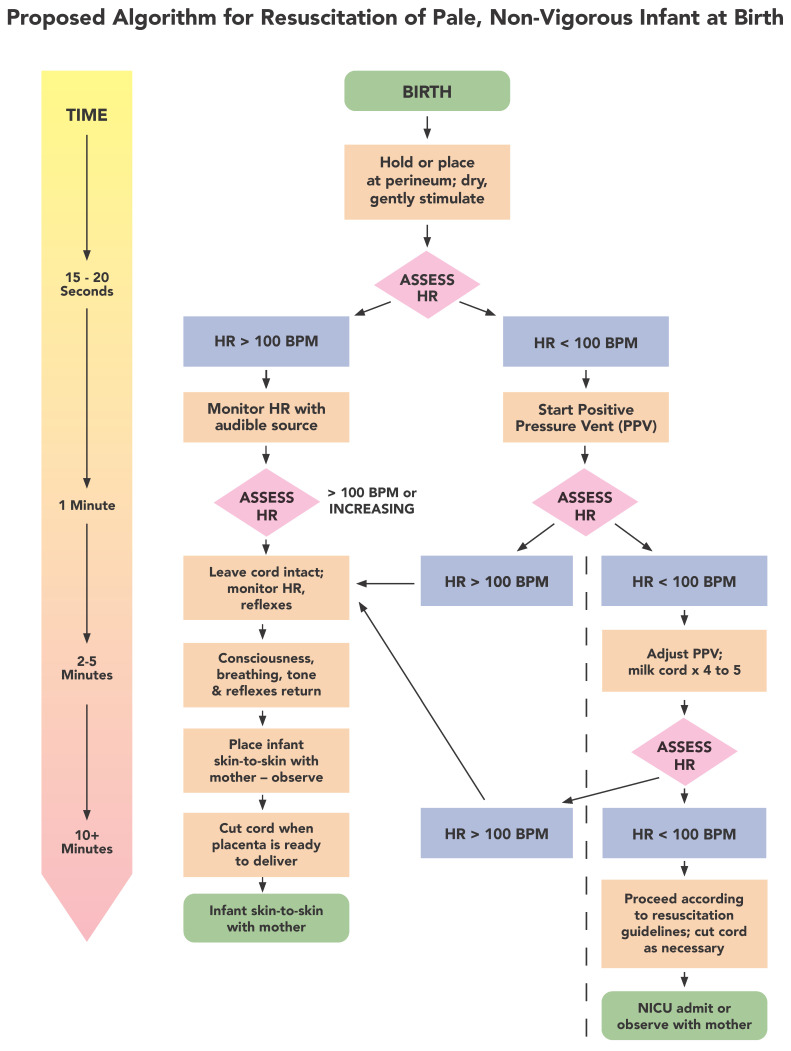
Proposed Algorithm for Resuscitation of Pale, Non-Vigorous Infants at Birth. HR: Heart Rate, BPM: Beats per Minute PPV: Positive Pressure Vent, NICU: Neonatal Intensive Care Unit; Timeline on left does not apply to items on right of the dotted line after 2–5 minutes.

**Table 1 children-10-00383-t001:** Commonalities between newborn cases of Sudden Cardiac Asystole.

	Mercer ^a^	Mercer ^b^	Ment ^a^	Ment ^b^	Ancora ^a^	Ancora ^b^	Cesari ^a^
AP heart rate	Normal	normal	normal	normal	normal	normal	normal
Birth weight (kg)	4.8	4.2	4.4	4.3	3.7	na	3.8
Labor h (2nd stage)	8 (2)	9 (1)	(1)	na	15	13	(1.5)
Head-body time (mins)	6	5–10	4.5	na	2	3	<5
Cord clamping	ICC	Before birth	ICC	ICC	ICC	ICC	ICC
1st heartbeat (mins)	23	18	25	12	7	13	None
Cord pH (art)	na	7.11	7.16	7.05	7.16	na	7.08
pH < 60 min	6.9	6.72	6.76	6.92	6.8	7.09	6.68
Hypotension	S	S	na	S	M	S	died
HIE	S	S	S	S	S	S	died
Died at, or outcome	2 mo	CP, DD	1 week	2.5 h	Mild DD	Severe CP, DD	Died at birth

a = first case, b = second case by same author; AP = antepartum (labor); art = arterial; DD = developmental delay; h = hours; HIE = hypoxic–ischemic encephalopathy; kg = kilogram; Ment = Menticoglou; mins = minutes; m = moderate; na = not available; S = severe. Mercer [1]; Menticoglou [2]; Ancora [3]; Cesari [4].

**Table 2 children-10-00383-t002:** Characteristics of two pathways of the vagus nerve.

Properties	Ventral Vagus	Dorsal Vagus
Origin in brainstem	Nucleus Ambiguus	Dorsal Motor Nucleus of the Vagus
Organs affected	Above the diaphragm	Below the Diaphragm
Evolutionary timeline	200 million years ago, at emergence of mammals	500 million years ago for all vertebrates
Effects on heart	Conveys respiratory rhythm to the heart, slows heart rate to normal	Usually no effect unless life threat — then slows or stops heart
Myelination	Myelinated	Unmyelinated
Physical signs without stress	Calm, open for sociality and connection, good vagal tone (RSA)	Supports vegetative processes; quiet
Vagal tone with extreme stress	Depressed	Activated
Physical signs with extreme stress	Low tone, depressed	Syncope, dissociation, defecation, bradycardia, asystole

## Data Availability

No new data created.

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
