# Peer review of "Cardiac Asystole at Birth Re-Visited: Effects of Acute Hypovolemic Shock"

_children, 2023, doi:10.3390/children10020383_

Round 1

Reviewer 1 Report

The reference number (25) for MINVI trial is incorrect

The limited number of patients case series with shoulder dystocia and nuchal cord weakly supports the hypothesis though physiological plausible. More evidence either from animal or human studies would be supportive. 

Author Response

Thank you for your review. We have fixed the reference numbers for the MINVI trial.

We have added additional text and references relating to the process of resuscitating pale non-vigorous infants. 

Reviewer 2 Report

This is an interesting manuscript giving pathophysiological explanations to unexpected cardiac arrest of newborn infants immediately after delivery. It may help to adapt postnatal management in compromised newborns after birth by applying delayed cord clamping or milking. However, the authors  should more clearly specify for which group of newborns they recommend delayed cord clamping. In my understanding they do not recommend delayed cord clamping in general interm newborns as this is the case with preterm delivery?

Chapter 5 desribes the autonomic nervous system and its implications in extensive detail. This should be shortended substantially to keep the focus on the core message of the manuscript.

Chapter 7, 2nd paragraph: The reference [25] cited for the MINVI trial seems to be wrong.

Author Response

Thank you for your review. We do support delayed cord clamping for all infants. The group of newborns for whom this paper was written are the ones born pale and non-vigorous. Even in institutions where delaying cord clamping is routine, these infants needing resuscitation still experience immediate cord clamping putting them at risk for ischemia, hypoxia, inflammation and death.

We have shortened Chapter 5 somewhat. However, we feel that how the autonomic nervous system functions under differing physiological situations,  especially hemorrhage-like circumstances, is part of the core message.  Sections 5.1.4, 5.2, and 5.3 contain the main point of the article - that it is the action of the dorsal vagal complex that actually stops the heart. Any further suggestions on editing this section would be appreciated. 

Round 2

Reviewer 1 Report

Revised article is more concise and with more references on cord milking in non-vigorous infant. The proposed mechanism of benefit is better explained in the revised version.